# Patterns of Cerebrospinal Fluid Alzheimer’s Dementia Biomarkers in People Living with HIV: Cross-Sectional Study on Associated Factors According to Viral Control, Neurological Confounders and Neurocognition

**DOI:** 10.3390/v14040753

**Published:** 2022-04-04

**Authors:** Mattia Trunfio, Cristiana Atzori, Marta Pasquero, Alessandro Di Stefano, Daniela Vai, Marco Nigra, Daniele Imperiale, Stefano Bonora, Giovanni Di Perri, Andrea Calcagno

**Affiliations:** 1Infectious Disease Unit, Department of Medical Sciences, University of Turin at Amedeo di Savoia Hospital, 10149 Torino, Italy; 2Neurology Unit, Maria Vittoria Hospital, 10144 Torino, Italy; 3Laboratory Medicine, Maria Vittoria Hospital, 10144 Torino, Italy

**Keywords:** HIV, Alzheimer’s dementia, cerebrospinal fluid, biomarkers, beta amyloid, tau, phosphorylated tau, neurocognitive disorders, neurodegenerative disorders, central nervous system infections

## Abstract

People living with HIV (PLWH) age with an excess burden of comorbidities that may increase the incidence of age-related complications. There is controversy surrounding the hypothesis that HIV can accelerate neurodegeneration and Alzheimer’s dementia (AD). We performed a retrospective study to analyze the distribution of cerebrospinal fluid (CSF) AD biomarkers (beta amyloid 1–42 fragment, tau, and phosphorylated tau) in adult PLWH (on cART with undetectable viremia, *n* = 136, with detectable viremia, *n* = 121, and with central nervous system CNS disorders regardless of viremia, *n* = 72) who underwent a lumbar puncture between 2008 to 2018; HIV-negative controls with AD were included (*n* = 84). Five subjects (1.5%) presented CSF biomarkers that were compatible with AD: one was diagnosed with AD, whereas the others showed HIV encephalitis, multiple sclerosis, cryptococcal meningitis, and neurotoxoplasmosis. Regardless of confounders, 79.6% of study participants presented normal CSF AD biomarkers. Isolated abnormalities in CSF beta amyloid 1–42 (7.9%) and tau (10.9%) were associated with age, biomarkers of intrathecal injury, and inflammation, although no HIV-specific feature was associated with abnormal CSF patterns. CSF levels of AD biomarkers very poorly overlapped between HIV-positive clinical categories and AD controls. Despite the correlations with neurocognitive performance, the inter-relationship between amyloid and tau proteins in PLWH seem to differ from that observed in AD subjects; the main driver of the isolated increase in tau seems represented by non-specific CNS inflammation, whereas the mechanisms underlying isolated amyloid consumption remain unclear.

## 1. Introduction

With the increasing life expectancy of people living with HIV (PLWH), the diagnosis and management of co-morbidities is crucial in HIV care; indeed, health-adjusted life expectancy is shorter in females and males living with HIV [1,2]. In countries where HIV care is extensively provided, such as the USA, PLWH aged 65 years and above doubled in number between 2013 to 2018, and by 2030, they are estimated to represent over 25% of all PLWH on combination antiretroviral therapy (cART) [3]. Similarly, by 2030, 73% of PLWH in Europe are projected to be 50 years of age or older [4].

Despite the fact that, just few decades ago, HIV-associated dementia was a common, severe diagnosis in AIDS patients, nowadays, age-related neurodegenerative disorders in ageing PLWH may be as common as in the general population or, as some evidence suggests, they may be even more prevalent [5,6,7]. For instance, the burden of age-related comorbidities associated with a higher risk of Alzheimer’s dementia (AD) is increased in PLWH compared to age-matched HIV-negative subjects, since, despite being controlled by cART, HIV infection can accelerate or accentuate the pathological processes underlying frailty and senescence [8]. Recent findings described an overall higher incidence of dementia among PLWH compared to HIV-negative subjects (incidence rate ratio of 1.80 after adjusting for sociodemographic and clinical factors), despite a decreasing trend in both populations during the analyzed period (2000–2016) [5]. Unfortunately, dementia diagnoses covered also other types of disorders that are different from AD, so direct inferences could not be easily drawn for this specific disease.

In line with the recent hypothesis of an infectious trigger in AD pathogenesis [9], in vitro evidence suggests a direct role of HIV, or of some of its components (such as tat ang gag proteins), in favoring amyloid beta misfolding and plaque aggregation, as well as in promoting neuronal injury, tauopathies, and the hyperphosphorylation of tau proteins [10,11,12]. All these processes are the pathological hallmarks of AD, and they are targets of ongoing studies on therapeutic strategies. Similarly, the aetiopathogenesis of HIV-related neurocognitive disorders has not been completely clarified, and some authors have described clinical and pathological similarities with AD and other neurodegenerative dementias [8,13]. Some of these findings have also been attributed to antiretrovirals’ effects rather than to HIV itself [14,15]. Nevertheless, very few cases of AD among PLWH have been reported in the cART era so far [16,17,18,19]. A recent multi-centric case series observed an overall AD prevalence of 0.44/100,000 among more than 9000 PLWH; however, regardless of viral control, age, or potential AD risk factors [20], no in vivo study observed compelling evidence in support of accelerated or accentuated AD-like phenomena that was directly attributable to HIV infection. Moreover, other data pointed towards the role that some antiretrovirals have in counteracting the pathological dynamics associated with AD development [14,21]. Furthermore, imaging and biomarkers studies described negative findings, and a different pattern of cerebrospinal fluid (CSF) biomarkers in very selected and limited groups of HIV-positive subjects, compared to those characterizing AD [22,23,24,25].

To date, no single reliable diagnostic tool is available for differentiating AD from HIV-related neurological disorders, but the therapeutic implications of this distinction are vital. Outside neuroAIDS research, data endorsing a significant viral contribution in AD are increasing; the primarily identified viruses are *Herpesviridae* (HSV-1, HHV-6, and -7), and the antimicrobial role of amyloid beta against viruses, bacteria, and fungi invading the central nervous system (CNS) has been recently described [9,26].

Given the complexity of host-viral interactions and the high prevalence of comorbidities, several confounding factors may emerge when assessing trends and values of AD CSF biomarkers in PLWH. In light of conflicting evidence, we described CSF AD biomarker levels and their current combinations in three common clinical categories of PLWH (patients on cART and undetectable viremia, patients with detectable viremia for any reason, and patients with CNS disorders regardless of viremia), and we assessed whether several HIV-related and unrelated characteristics may differentially associate with CSF biomarkers patterns in each clinical group.

## 2. Materials and Methods

We performed a retrospective prevalence study to describe the distribution of CSF AD biomarkers in adult PLWH who underwent a lumbar puncture (LP) for clinical or research reasons from January 2008 to October 2018 at our Infectious Diseases Unit (Amedeo di Savoia Hospital, Turin, Italy). We also assessed factors potentially associated with the CSF biomarkers’ combinations. We included patients representing the current three most common clinical categories in PLWH:(1)patients on cART for at least 6 months, undetectable plasma viremia (defined as plasma HIV-RNA < 50 cp/mL), and no HIV-unrelated neurological conditions,(2)patients with detectable plasma viremia who are either on cART or not, and who have no HIV-unrelated neurological conditions,(3)patients with any confounding neurological conditions not directly linked to HIV infection regardless of plasma suppression.

Neurological conditions deemed as HIV-related were symptomatic or asymptomatic CSF escape (defined as detectable CSF HIV-RNA with undetectable plasma HIV-RNA or CSF HIV-RNA 0.5 Log10 higher than plasma viremia if the latter was detectable, as per previously recommended definitions [27]), HIV encephalitis, HIV-related neurocognitive disorders, neurological complaints without classified aetiologies (e.g., headache, self-reported neurocognitive deterioration without compatible neurocognitive testing results), and isolated brain MRI white matter abnormalities. CNS infections, either due to opportunistic or non-opportunistic pathogens, cancers, cerebrovascular events, and demyelinating or autoimmune CNS disorders were coded as HIV-unrelated neurological confounding.

Demographic, clinical and viro-immunological data were recorded, as well as the results of CSF analysis. LP was performed between 10:00 and 12:00 am to reduce intra- and inter-subject daily variability in CSF circulation and drainage; once collected, the CSF samples were immediately sent for biomarker measurement, without intermediate stock or freezing steps. CSF total tau (tau), 181-phosphorylated tau (ptau), and β-amyloid 1–42 fragments (BA42) were measured using immunoenzymatic methods (Innogenetics, Ghent, Belgium, EU) with limits of detection of 87, 15, and 87 pg/mL, respectively. Neopterin and S100β were measured through validated ELISA assays (DRG Diagnostics, Marnurg, Germany, and DIAMETRA Srl, Spello, Italy, respectively). Normality thresholds were as follows: tau < 300 pg/mL (in patients aged ≤50 years), <450 pg/mL (in patients aged 51–70 years old), and <500 pg/mL in older patients; ptau < 61 pg/mL; BA42 > 500 pg/mL; neopterin < 1.5 ng/mL; S100β < 380 pg/mL. The CSF-to-serum albumin ratio (CSAR) was calculated as CSF albumin (mg/L)/serum albumin (g/L), and used to assess blood–brain barrier integrity according to age-adjusted reibergrams (normal if <6.5 in patients aged <40 years and <8 in older patients) [28].

Patients were classified into CSF biomarker patterns according to CSF tau and BA42 combinations as follows:Pattern (A) when both the biomarkers were within the normal range;Pattern (B) when BA42 was low while tau was normal;Pattern (C) when BA42 was normal while tau was elevated;Pattern (D) when BA42 was low, and tau was elevated (CSF pattern compatible with AD).

All the study subjects underwent brain imaging through 1.5 tesla magnetic resonance imaging (MRI) with standard T1-, T2-, diffusion- and FLAIR-weighted sequences (contrast enhancement was administered as per clinical indications only).

Within ±6 months from the LP, a subgroup of patients underwent neurocognitive evaluation based on 16 tests assessing 6 cognitive domains: Trail Making test parts A and B, Stroop color test, Digit Span forward and backward, Digit Symbol, Corsi and Disyllabic Words Serial Repetition tests, Free and Cued Selective Reminding test, Story Recall, Frontal Assessment battery, immediate and delayed recall of the Rey–Osterrieth complex figure, phonemic, semantic, and alternate verbal fluency, and Grooved Pegboard for the dominant and non-dominant hand. For study purposes, no classification of HIV-associated neurocognitive impairment was adopted; patients were categorized as symptomatically neurocognitively impaired (NCI) in cases of at least two domains below 1 standard deviation compared with normative data in the presence of a self-reported deficit (by the Lawton’s Instrumental Activities of Daily Living scale), or a deficit identified by physicians or the caregiver if any, in everyday functioning [29].

Since, as previously described, the Apolipoprotein E (APOE) ε4 genotype could play a relevant role in CSF beta amyloid metabolism in PLWH [30], and different HIV-1 subtypes may show differential neurotropism and CNS virulence [31], data on APOE genotyping (DiaPlexC Apolipoprotein E Genotyping kit, SolGent Co., Ltd., Daejeon, Korea), and HIV-RNA subtypes (ViroSeq HIV-1 Genotyping System, Abbott, Chicago, IL, USA) were retrieved for all subjects with the available data.

An HIV-negative control group was enrolled and was represented by all the adult patients diagnosed with AD, and with available LP analysis at the Neurology unit, Maria Vittoria Hospital, Turin, Italy, during the same period of inclusion as HIV-positive cases. CSF analysis of the controls was performed with the same assays and procedures adopted for HIV-positive patients. The diagnosis of AD was based according to 2018 NIA-AA guidelines [32].

The study was conducted according to the guidelines of the Declaration of Helsinki and approved by the Ethics Committee of San Luigi Hospital, Orbassano (“Prospective study on predictors of neurocognitive decline in HIV-positive patients” PRODIN, protocol code 103/2015, approved on 22 June 2015).

Data were analyzed through nonparametric tests (Spearman’s rank correlations, Kruskal–Wallis test, Mann–Whitney test) to reduce type I errors due to multiple testing, and to limit the effect of potential outliers. Continuous variables were reported as medians (interquartile range) and categorical variables were reported as absolute numbers (proportion). The study size was determined by the available CSF samples that were analyzed a posteriori. Selection biases were limited by the long recruitment period and the consecutive testing of any subjects undergoing a LP. Missing data were approached using the deletion method. Binary logistic regressions were run after including variables that showed univariate *p*-values < 0.1. Data analysis was performed using SPSS v27 (IBM Corp., Armonk, NY, USA). Data reporting follows the STROBE checklist.

## 3. Results

### 3.1. Study Population

Overall, 442 HIV-positive subjects underwent a LP during the study period; of these, 21 did not meet the eligibility criteria, and 92 did not have their AD biomarkers measured due to denied consent, a missing request, or a lack of CSF. Eventually, 329 HIV-positive patients were included. One hundred and thirty-six and 121 patients had plasma HIV-RNA< and >50 cp/mL, respectively; furthermore, 72 patients presented at least one neurological condition regardless of peripheral viral suppression.

Among undetectable patients, 122 belonged to group A (89.7%), 9 to B (6.6%) and 4 to C (2.9%); only one patient had a CSF profile that was compatible with AD (0.7%). Among patients with detectable viremia, 97 belonged to group A (80.2%), 9 to B (7.4%), and 14 to C (11.6%), with only one subject having a CSF profile that was compatible with AD (0.8%). Forty-three patients with neurological confounding belonged to group A (59.7%), 8 to B (11.1%), 18 to C (25.0%), and 3 to D (4.2%). Of these, 38 subjects (52.8%) had at least one CNS opportunistic infection (17 PML, 10 neurotoxoplasmosis, 6 cryptococcal, and 5 tuberculosis meningitis, 4 CMV encephalitis), 8 had non-opportunistic CNS infections (11.1%; 6 neurosyphilis, and 1 HSV-1 and VZV encephalitis), 6 had primary CNS B-cells lymphoma (8.3%), and another 22 presented other non-infectious disorders (30.6%; 1 neurosarcoidosis, 1 optic neuritis and 1 optic neuromyelitis, 3 multiple sclerosis, 6 inflammatory myelitis, 2 Guillain–Barre syndrome, 7 vascular events, and 1 neuropsychiatric manifestations during sepsis).

Overall, 239 (72.6%), 183 (55.6%), and 160 (48.6%) HIV-positive subjects had data on APOE genotyping, CSF HIV subtype, and neurocognitive evaluation. The demographics, clinical, and viro-immunological characteristics of the three groups are summarized in Table 1. The spectrum of the four CSF patterns of AD biomarkers in the HIV-positive clinical groups is depicted in Figure 1. More specifically, a different distribution of AD biomarkers was observed: CSF pattern A was similar among undetectable (89.7%) and detectable patients (80.2%), and was more represented in both the categories compared to confounding (59.7%; *p* < 0.0005 and *p* = 0.001, respectively). Pattern C was more predominant in confounding compared to detectable (25.0% vs. 2.9%, *p* = 0.004) and undetectable (vs. 11.6%, *p* < 0.0005), and in the former compared to the latter (*p* = 0.027). Patterns B and D were similarly represented among the three groups (6.6% vs. 7.4% vs. 11.1%, *p* = 0.507 and 0.6% vs. 0.8% vs. 4.2%, *p* = 0.116; see Figure 1).

Eighty-four HIV-negative patients diagnosed with AD were included as control group. Forty-four (52.4%) were male and the median age was higher than the HIV-positive counterpart (70 years (60–76), *p* < 0.001). Applying the same assay for CSF biomarker analysis, tau (681.7 pg/mL [545.6–984.3]) and ptau (81.5 pg/mL [67.5–103.4]) were extremely higher compared to the three HIV-positive groups (*p* < 0.001 for both), whereas BA42 was significantly lower (384.3 [308.8–431.3], *p* < 0.001, as shown in Figure 2; the median values for the HIV-positive study population are reported in Table 1 and Figure 2).

In the control group, CSF tau correlated with ptau (rho 0.543, *p* < 0.0005), and not with BA42 (rho −0.125, *p* = 0.26), nor age (rho −0.057, *p* = 0.60). CSF ptau and BA42 showed a very mild correlation with each other (rho 0.214, *p* = 0.051) and none with age (rho 0.007, *p* = 0.95 and rho −0.103, *p* = 0.35, respectively). Interestingly, age did not correlate with any of the AD biomarker CSF levels in the HIV-negative control group, and only CSF tau levels mildly correlated with age in the whole HIV+ study population (see Appendix A).

### 3.2. HIV-Positive Subjects with Undetectable Viremia

Among the nine subjects with isolated low BA42, six had neurocognitive impairment (five asymptomatic and one symptomatic), two cases had no neurological/neurocognitive issue (one presenting white matter abnormalities), and the last had asymptomatic CSF escape.

Half of the four cases with CSF pattern C were related to CSF escape (one symptomatic and one asymptomatic), whereas the other half was detected in neurocognitive impaired subjects (one symptomatic and one asymptomatic).

We assessed through univariate binary logistic regression potential factors associated with the odds of detecting a CSF pattern of type B or C among the 136 subjects with undetectable viremia. Univariate analysis showed increased odds of presenting isolated low BA42 as age of subjects increased (OR 1.07 [1.002–1.15], *p* = 0.048), and no other demographic, clinical, or viro-immunological parameter was significantly associated with CSF pattern B in this clinical group (Appendix A).

Length of HIV infection (OR 1.017 [1.003–1.0032], *p* = 0.020) and CSF cell count (OR 1.84 [1.07–3.18], *p* = 0.028) were the only factors associated with increased odds of detecting a CSF pattern C among undetectable subjects during univariate analysis, but no factor was independently associated with isolated increased tau during multivariate analysis in this clinical group (Appendix A). Including HIV subtype, APOE genotype, or both did not change the findings of the multivariate analysis (data not shown).

In patients with undetectable viremia, tau correlated with ptau (rho 0.679, *p* < 0.0005), BA42 (rho 0.351, *p* < 0.0005), CSAR (rho 0.313, *p* < 0.0005), age (rho 0.472, *p* < 0.0005; Appendix A), CSF cells (rho 0.280, *p* = 0.001), and proteins (rho 0.368, *p* < 0.0005; see Figure 3). Similarly, ptau correlated with BA42 (rho 0.470, *p* < 0.0005), CSAR (rho 0.399, *p* < 0.0005), age (rho 0.249, *p* = 0.004; Appendix A), and CSF proteins (rho 0.410, *p* < 0.0005). BA42 correlated with CSAR (rho 0.349, *p* < 0.0005) and CSF proteins only (rho 0.388, *p* < 0.0005; see Figure 3 that reports the correlations with the highest rho). None of the three CSF AD biomarkers correlated with any of the viro-immunological parameters.

Eighty-one (59.6%) patients underwent neurocognitive assessment. CSF tau correlated with age-, sex- and education-adjusted scores during the Free and Cued selective reminding test delayed part (rho −0.343, *p* = 0.011), Frontal assessment battery (rho −0.325, *p* = 0.009), Digit symbol (rho −0.332, *p* = 0.012), and Grooved Pegboard for dominant (rho 0.334, *p* = 0.014) and non-dominant hand (rho 0.318, *p* = 0.022), as shown in Figure 4. CSF tau also correlated with the overall score of two screening tests for neurocognitive impairment: International HIV Dementia Scale (IHDS; rho −0.254, *p* = 0.022), and Mini-Addenbrooke’s Cognitive Examination test (MACE; rho −0.499, *p* = 0.008). Similarly, CSF ptau correlated with semantic verbal fluency (rho −0.786, *p* = 0.036; Figure 4) and MACE (rho −0.463, *p* = 0.015). BA42 weakly correlated with the adjusted score of Trail Making test part A only (rho 0.291, *p* = 0.012; Figure 4). The interpretation of the correlation between CSF biomarkers and neurocognitive tests is depicted in Figure 4; more specifically, the effect size of each statistically significant correlation has been reported by cross tabulating the biomarker of interest with the main cognitive domain of whose tests the scores were found to correlate with.

### 3.3. HIV-Positive Subjects with Detectable Viremia

Of the 9 subjects with isolated low BA42, 2 had symptomatic CSF escape, 1 had primary HIV infection, 1 had HIV encephalitis, 1 had symptomatic neurocognitive impairment, and 4 had no neurocognitive/neurological issue (half presented white matter abnormalities).

Of the 14 subjects with isolated increased tau, one had primary HIV infection, four had neurological complaints in the presence of a high CSF HIV-RNA (>100.000 cp/mL) but no CSF escape, six had no neurocognitive/neurological issue (three were in the presence of a high CSF HIV-RNA), and three had neurocognitive impairment (one asymptomatic).

We assessed through univariate binary logistic regression potential factors associated with the odds of detecting a CSF pattern of type B or C among the 122 subjects with detectable viremia. Univariate analysis showed increased odds of presenting isolated low BA42 as age (OR 1.13 [1.05–1.22], *p* = 0.001), CSF neopterin (OR 1.24 [1.06–1.44], *p* = 0.005), as Tourtelotte index increased (OR 1.03 [1.01–1.05], *p* = 0.004), and as CSF glucose decreased (OR 0.89 [0.81–0.97], *p* = 0.012). With the multivariate model, which also included CSF proteins (univariate *p*-value 0.062), increased age (aOR 1.16 [1.03–1.31], *p* = 0.017) and CSF glucose (aOR 0.85 [0.75–0.97], *p* = 0.019) were independently associated with increased and decreased odds of detecting CSF pattern B, respectively (Appendix A).

Increasing values in CSF neopterin (OR 1.34 [1.13–1.58], *p* = 0.001), Tourtelotte index (OR 1.02 [1.004–1.04], *p* = 0.015), CSF cells (OR 1.04 [1.006–1.07], *p* = 0.020) and proteins (OR 1.04 [1.02–1.07], *p* = 0.002) were all associated with increased odds of detecting an isolated high tau; at multivariate, only CSF neopterin was independently associated with the odds of finding CSF pattern C (aOR 1.47 [1.10–1.96], *p* = 0.009; Appendix A). Including either CSF HIV subtype, APOE genotyping, or both in multivariate analysis did not change the findings (data not shown).

In patients with detectable viremia, tau correlated with CSF BA42 (rho 0.503, *p* < 0.0005), ptau (rho 0.672, *p* < 0.0005), neopterin (rho 0.298, *p* = 0.001), cells (rho 0.283, *p* = 0.002) and proteins (rho 0.219, *p* = 0.017; Figure 5). BA42 correlated with ptau (rho 0.541, *p* < 0.0005) and Tourtelotte index (rho −0.207, *p* = 0.023; Figure 5 reports the correlations with the highest rho). None of the three CSF AD biomarkers correlated with any of the viro-immunological parameters nor with age.

Fifty-four patients (44.6%) underwent neurocognitive assessment. CSF tau correlated with the adjusted score of the Free and Cued Selective reminding test delayed part (rho −0.383, *p* = 0.044) and IHDS (rho −0.410, *p* = 0.002; Figure 4). CSF ptau correlated with the adjusted score of phonemic verbal fluency (rho −0.285, *p* = 0.043) and the immediate copy of the Rey–Osterrieth complex figure (rho −0.303, *p* = 0.031; Figure 4). BA42 correlated with the adjusted score of the Disyllabic Words Serial Repetition test (rho −0.424, *p* = 0.001), phonemic verbal fluency (rho −0.400, *p* = 0.003), and Grooved Pegboard for non-dominant hand (rho 0.411, *p* = 0.022; Figure 4).

### 3.4. HIV-Positive Subjects with Central Nervous System Confounding

Through univariate binary logistic regression, we assessed potential factors associated with the odds of detecting a CSF pattern of type B or C among the 72 subjects with HIV-unrelated CNS disorders. Univariate analysis showed a reduced probability of detecting a CSF pattern B in Caucasians (OR 0.17 [0.029–0.99], *p* = 0.050) and males (OR 0.12 [0.023–0.60], *p* = 0.010). Using multivariate analysis, including the HIV acquisition route (univariate *p*-value for previous drug users 0.060) and CSAR (univariate *p*-value 0.078), no parameter independently predicted differential odds of detecting CSF pattern B in this clinical group (Appendix A). Similarly, the odds of detecting pattern C differed according to age (OR 0.91 [0.85–0.97], *p* = 0.007) and IgG intrathecal synthesis (OR 1.04 [1.007–1.07], *p* = 0.017), but no variable confirmed an independent association with the odds of an isolated increased tau in patients with CNS confounding (Appendix A).

CSF tau correlated with ptau (rho 0.479, *p* < 0.0005), but not with BA42, and with current CD4+ T-cells count (rho −0.352, *p* = 0.003), CSF HIV-RNA (rho 0.348, *p* = 0.003), neopterin (rho 0.369, *p* = 0.009), length of HIV infection (rho 0.309, *p* = 0.009), and length of viral suppression (rho −0.457, *p* = 0.022). Instead, CSF ptau did correlate with BA42 (rho 0.623, *p* < 0.0005), as well as with the length of viral suppression (rho −0.417, *p* = 0.038). BA42 did not correlate with any assessed variable, but for ptau. None of the three CSF AD biomarkers correlated with age.

Twenty-five patients (34.7%) underwent neurocognitive evaluation. None of the three CSF biomarkers correlated with any score of the 16 tests or with the two screenings administered (as shown in Figure 4).

### 3.5. HIV-Positive Subjects with Cerebrospinal Fluid Biomarkers Profile Compatible with Alzheimer’s Dementia

Overall, 5 patients (1.5%) presented a CSF biomarker profile that was compatible with, and typical of AD; out of those five patients, 1 patient had undetectable viremia, another had detectable viremia, and 3 patients had CNS confounding. Only the first was diagnosed with AD in 2018 and in depth previously described [20].

The 41 year-old man with detectable viremia suffered from HIV encephalitis with acute clinical onset in the advanced stages of HIV infection (fever, headache, confusion, dizziness, tremors). The brain MRI showed diffuse and symmetrical T2-weighted hyperintensities within the white matter of centrum semiovale, corona radiata, internal capsula and the brain stem. After the introduction of cART, he ameliorated, showing normalization of imaging, symptoms, CSF BA42, and reduction of tau: 14 months after the acute episode, CSF BA42 moved from 365.0 to 1029.0, tau from 811.0 to 515.0 pg/mL, and CSF HIV-RNA from 375,412 to 227 cp/mL.

The last three subjects suffered from the first flare of new-onset multiple sclerosis (24 year-old male), cryptococcal meningitis (41 year-old female) and neurotoxoplasmosis (49 year-old male), respectively. All of them observed a normalization of CSF biomarkers after the introduction of the respective specific treatments at a mean follow-up of 3 months.

Therefore, the overall prevalence of actual AD among our cohort was 0.3%.

## 4. Discussion

Regardless of neurocognitive impairment, antiretrovirals, viral replication, immunological status, or CNS confounding, most of the study participants presented normal CSF AD biomarkers (79.6%, and up to 89.7% in patients with undetectable viremia). No single HIV-related variable was associated with abnormal CSF biomarker patterns. More importantly, CSF median levels of AD biomarkers very poorly overlapped between HIV-positive clinical categories and HIV-negative AD controls, suggesting a reliable power of discrimination between HIV-related neurological processes and AD.

Despite several hypothetical parallelism between AD pathogenesis and HIV-related CNS phenomena, such as the role of chronic inflammation within CNS which was observed in both disorders [33,34,35], our findings are reassuring in terms of recent concerns of a hypothetical pro-amyloidogenic role of HIV and of antiretroviral therapy [10,11,12,14,15,36]. Our study was not designed to assess accelerated or accentuated neurodegeneration in PLWH, but if it takes place, these processes seem to lead to tau and amyloid protein aggregation processes that are different in magnitude, and have different mechanisms, from those detected by validated combinations of CSF biomarkers for AD. Indeed, correlations among tau proteins and BA42 are not clearly defined and are variable in HIV-negative subjects suffering from AD: increasing values of tau and ptau were correlated with a decrease in normal or high values of BA42 [37]; among our controls, no correlation was detected between tau proteins and BA42. On the contrary, among our study HIV-positive population, increasing values of tau and ptau were detected along increasing values of BA42 in every category except in CNS confounding, as previously observed [38,39]. Lastly, the biomarker alterations observed in patients with an AD-like CSF pattern were reversible in all the cases where an alternative diagnosis was found. In line with what our findings suggest, amyloid plaques found in PLWH possess a different configuration and localization compared to those characterizing AD [40], and imaging studies tailored for AD mechanisms (such as brain tau- and amyloid-PET) have provided inconclusive results in PLWH, with or without neurocognitive impairment [22,23,24,25].

Isolated low BA42, which was uncommonly found in the overall population (7.9%), as in each clinical category, did not differ according to CSF viral replication, cART administration nor the presence of other neurological infections or disorders. After excluding neurological confounding, we observed an association between isolated low BA42 and increasing age in both detectable and undetectable patients. This may primarily reflect physiological changes in BA metabolism with ageing, but also the fact that age can behave as a proxy of the length of continuous exposure of the brain to HIV proteins. Molecular mechanisms triggering amyloid deposition and their associated toxicity are not yet completely clarified, but over the last decades, emerging evidence highlighted that AB plays a role as innate immune protein that protects against fungal, bacterial, and viral infections [9,26]. As for other CNS viral infections, there are several data describing how the amyloid precursor protein and amyloid fragments can react to HIV presence to protect against brain infection [9,12,41,42]; since the interaction between AB and HIV could be directly or indirectly mediated by HIV proteins, the amyloid alterations observed in PLWH may not differ between detectable and undetectable subjects, as Gag and Tat are produced even in the absence of replicating virions, and the observed increased odds of isolated BA42 consumption with increasing age may also be due to the increasing time of exposure to such retroviral proteins. Whether or not these may explain what we found, isolated low levels of CSF BA42 seems uncommon in PLWH, but still clinically relevant, since in our population, as in previous others [43], lower CSF levels of BA42 correlated with worse cognitive outcomes.

The main limitation in assessing CSF BA42 is represented by the cross-sectional design. Longitudinal studies could be more accurate in deciphering at which point the subject is at with regard to the hypothetical evolution of BA42 levels in response to HIV infection, as it is likely that, after an infectious trigger, CSF concentrations of BA42 change over time, increasing and decreasing according to the type, amount, and duration of the trigger itself. In potential agreement with this, the association between normal/higher CSF glucose levels and lower odds of presenting isolated low BA42 in subjects with the replicating virus may identify in CSF glucose a more sensitive and long-lasting representative marker of intrathecal chronic and subtle inflammatory processes affecting BA metabolism (as already observed in other chronic CNS infections such as tuberculosis meningitis [44]). On the contrary, other biomarkers such as neopterin (marker of monocyte/macrophage activation only) or CSF cells (which do not correlate with cell activity in brain parenchyma) may not be specific or sensitive enough for alterations in BA metabolism when HIV is replicating. Secondly, we could not compare BA42 concentrations according to a single antiretroviral compound, but preliminary evidence may suggest that there is an impact of different compounds on BA brain metabolism in PLWH [14,15,36].

A mild contribution in increased total tau and ptau was given by CNS inflammatory infectious or non-infectious processes (10.9%). Indeed, we observed an increasing prevalence of isolated high CSF tau in the three clinical groups, with the highest being in those presenting CNS confounding conditions. Both in detectable and undetectable patients, an isolated increase in CSF tau (and ptau) was associated with common and aspecific markers of intrathecal inflammation: CSF cells and proteins, neopterin, and Tourtelotte index. Once again, the observed infrequent alterations in markers of tauopathies highlight aspecific mechanisms underlying the increase in tau and ptau, rather than HIV-related processes. In line with this finding, being markers of axonal injury, tau and ptau have been found to be elevated in several CNS infections during the acute flare [39,45,46]. Even more relevant than for BA42 alterations, the clinical counterpart of increased CSF concentrations of tau and ptau was represented by worse cognitive outcomes in several cognitive domains; further studies on the involvement of CSF tau proteins in the neurocognition of PLWH are warranted, as contrasting results have been reported in the literature to date [43,47]. Interestingly, CSF levels of tau and ptau (and not BA42) mildly positively correlated with age in the undetectable group only (as shown in Appendix A). These correlations were lost in the other groups, where the background of CSF replicating HIV and of other confounding factors may have altered the association and were not detected in the HIV-negative control group of subjects with AD; thus, the relationship between age and CSF levels of tau and ptau among subjects with undetectable viremia could be explained by a potential HIV-related legacy effect on neuronal injury and axonal damage. This legacy could be more prominent in older PLWH compared to the younger ones that have had a shorter period of untreated viral replication.

Strengths of this study are represented by the largest sample size and the most extensive assessment (clinical, imaging, biomarkers, and neurocognition data) compared with previous studies on CSF AD biomarkers in PLWH, by the novelty of grouping patients according to CSF patterns rather than analyzing single biomarkers, by the presence of a negative control group for AD, as well as by the length of HIV infection in the cohort (median 8 years), that enhanced the likelihood to detect chronic and subacute processes, if any, despite the cross-sectional design. Lastly, compared to previous smaller studies on AD biomarkers in PLWH, our cohort enrolled subjects that better represent the current prototypes of patients in care.

As with other limitations, we performed a retrospective snapshot without the capacity to describe or predict the eventual trajectory of the CSF AD biomarker levels of PLWH at the age of HIV-negative AD controls, which was indeed compatible with AD’s epidemiology, and therefore, about 20 years older than in our HIV-positive participants; this remains one of the main issues for directly comparing and translating AD biomarker levels and cut-off in PLWH. Furthermore, to date, findings describing CSF biomarkers of AD in PLWH should be considered of limited potential for a direct translation into routine clinical practice as the immunoassays adopted for measuring BA and tau proteins have never undergone a laboratory validation in the HIV-positive population.

## 5. Conclusions

In conclusion, clustering PLWH according to canonical patterns of CSF AD biomarkers did not detect any HIV-specific association; the inter-relationship between amyloid and tau proteins seem to differ from that observed in AD subjects and although the main driver of alterations in CSF tau proteins seem represented by non-specific inflammation within the CNS, the mechanisms underlying the rare isolated BA42 consumption remain unclear. Longitudinal studies are warranted to better depict long-term effects of HIV infection as well as of antiretroviral therapy on CNS amyloid and tau metabolism.

## Figures and Tables

**Figure 1 viruses-14-00753-f001:**
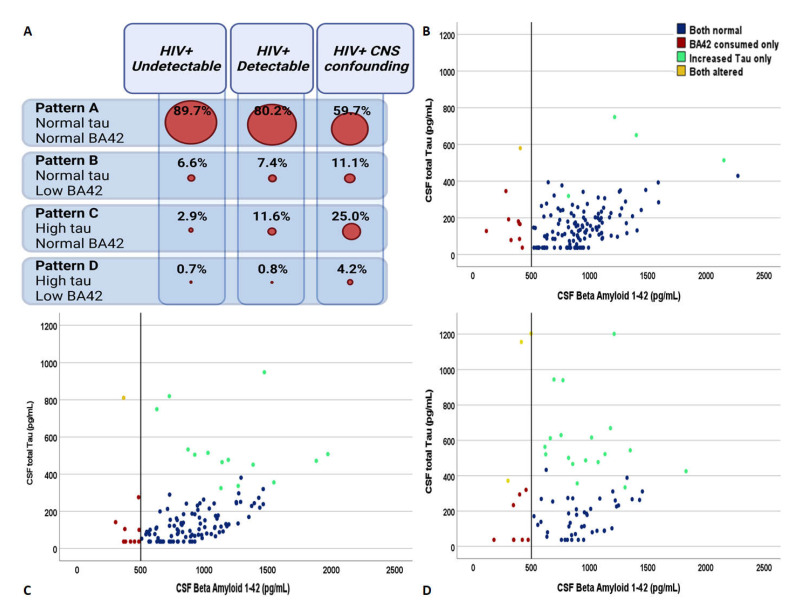
The distribution of the four patterns of cerebrospinal fluid biomarkers of Alzheimer’s dementia across the spectrum of HIV-related clinical conditions. Panel (**A**) shows the relative distribution of the four biomarker groups in the three clinical categories. Panel (**B**–**D**) describe the distribution of the study subjects identified according to the cross-combination of CSF total tau and beta amyloid 1–42 in the group of PLWH with undetectable plasma viremia, with detectable plasma viremia, and with neurological confounding conditions regardless of plasma viremia, respectively. Blue dots = subjects with CSF pattern A, red dots = subjects with CSF pattern B, green dots = subjects with CSF pattern C, and yellow dots = subjects with CSF pattern D.

**Figure 2 viruses-14-00753-f002:**
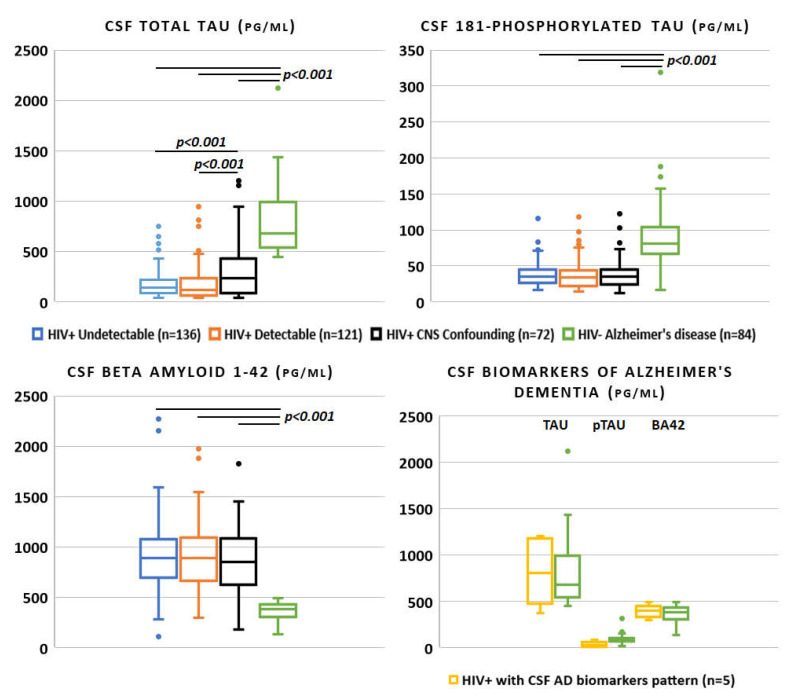
Alzheimer’s disease cerebrospinal fluid biomarkers across the spectrum of HIV infection and in negative controls (only *p*-values < 0.05 for the one-to-one group comparisons were reported).

**Figure 3 viruses-14-00753-f003:**
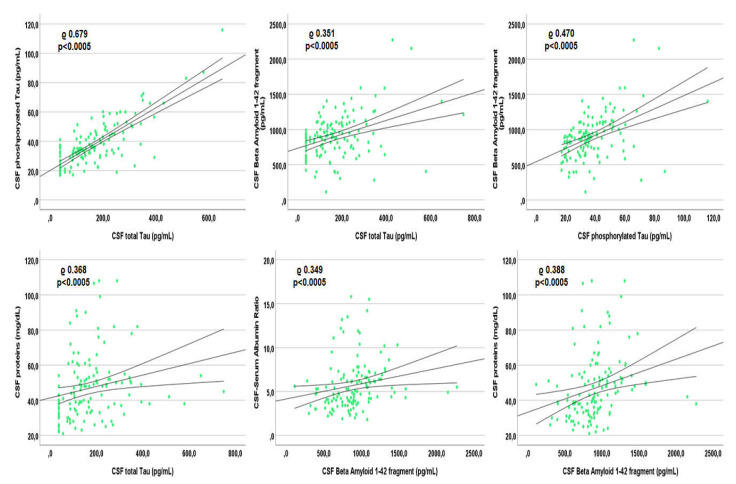
Spearman’s correlations between each other Alzheimer’s dementia biomarker and other markers within the cerebrospinal fluid of patients with undetectable plasma viremia.

**Figure 4 viruses-14-00753-f004:**
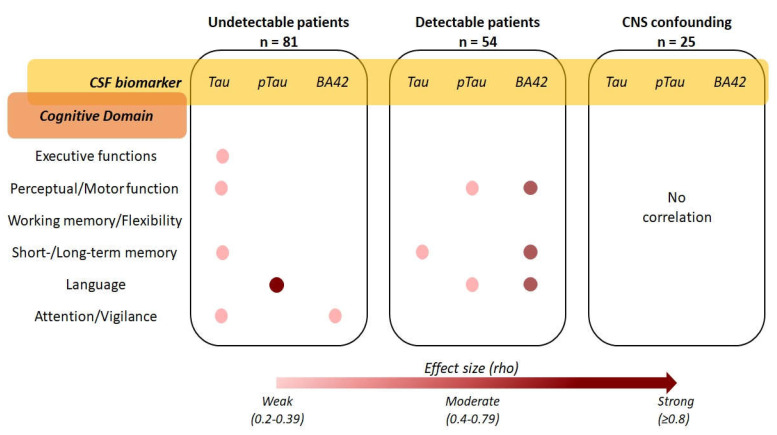
Strength and interpretation of the correlations between cerebrospinal fluid tau, ptau, and β Amyloid 1–42, and the results of the neurocognitive assessment in the three HIV-positive clinical groups (the exact rho values of the significant Spearman’s correlations are reported in the respective results sections).

**Figure 5 viruses-14-00753-f005:**
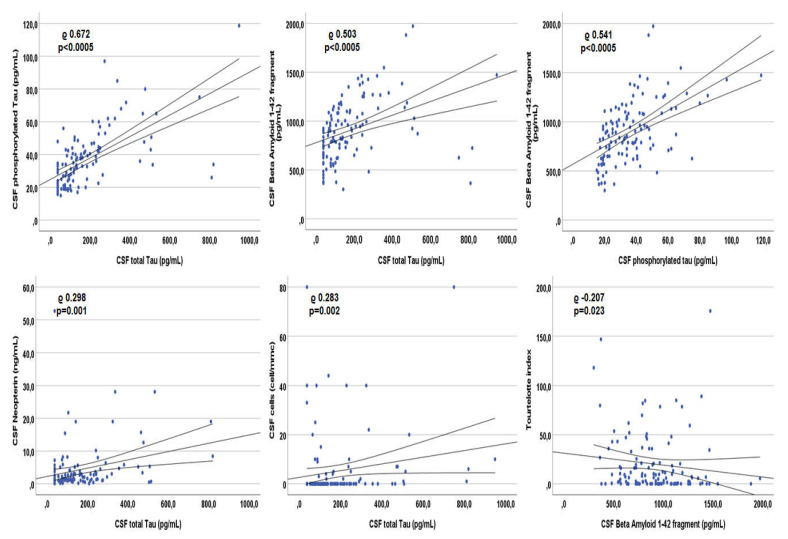
Spearman’s correlations between each other Alzheimer’s dementia biomarker and other markers within the cerebrospinal fluid of patients with detectable plasma viremia.

**Table 1 viruses-14-00753-t001:** Demographic, clinical, and viro-immunological characteristics of the study groups.

Parameter	Undetectable *n* = 136	Detectable *n* = 121	CNS Confounding *n* = 72	*p*-Value
**Caucasian, *n***	129 (94.8%)	99 (81.8%)	60 (83.3%)	0.003
**Age, years**	49 (43–56)	44 (37–51)	48 (37–56)	0.002
**Male sex, *n***	92 (67.6%)	90 (74.4%)	54 (75.0%)	0.350
**Exposure risk, *n***				0.388
**MSM**	53 (39.0%)	43 (35.5%)	30 (41.7%)	
**Heterosexual**	52 (38.2%)	45 (37.2%)	25 (34.7%)	
**Previous IDU**	31 (22.8%)	33 (27.3%)	17 (23.6%)	
**Positive HCV antibodies, *n***	46 (33.8%)	24 (19.8%)	9 (12.5%)	0.001
**Positive HBc antibodies, *n***	19 (13.9%)	23 (19.0%)	4 (5.6%)	0.665
**Plasma HIV-RNA, Log_10_ cp/mL**	1.3 (0.04–1.3)	5.2 (4.3–5.8)	2.6 (1.3–5.5)	NA
**Plasma HIV-RNA < 50 cp/mL, *n***	100 (100%)	-	28 (38.9%)	NA
**CSF HIV-RNA, Log_10_ cp/mL**	1.28 (0.04–1.38)	3.5 (2.4–4.3)	2.9 (1.4–4.2)	<0.0005
**CSF HIV-RNA < 50 cp/mL, *n***	113 (83.1%)	13 (10.7%)	22 (30.6%)	<0.0005
**CSF escape, *n***	27 (19.8%)	3 (2.5%)	18 (25.0%)	0.002
**Current CD4+ T-cells, cells/mmc**	450 (280–707)	78 (29–186)	118 (61–327)	<0.0005
**CD4+ T-cells count nadir, cells/mmc**	121 (43–233)	56 (21–130)	45 (19–130)	<0.001
**HIV subtype, *n***				0.480
**A**	0/73	0/106	1/60 (1.7%)	
**B**	55/73 (75.3%)	80/106 (75.5%)	39/60 (65.0%)	
**C**	0/73	3/106 (2.8%)	0/60	
**D**	0/73	0/106	1/60 (1.7%)	
**F**	5/73 (6.8%)	6/106 (5.7%)	13/60 (21.7%)	
**G**	5/73 (6.8%)	7/106 (6.6%)	3/60 (5.0%)	
**Recombinants**	8/73 (10.9%)	10/106 (9.4%)	3/60 (5.0%)	
**HIV infection, years**	14.0 (4.3–19.2)	1.3 (0.5–13.6)	5 (0.3–16.7)	<0.001
**On cART, *n***	100 (100%)	29 (23.9%)	45 (62.5%)	NA
**Time on current cART, months ***	15 (7–28)	4 (2–9)	6 (2–16)	NA
**Time of suppression, months ***	22 (8–71)	-	9 (6–56)	NA
**cART regimens, *n* ***				NA
**Dual**	24 (17.6%)	2/29 (6.9%)	7/45 (15.5%)	
**PI-NRTI**	41 (30.1%)	8/29 (27.6%)	13/45 (28.9%)	
**INI-NRTI**	30 (22.1%)	5/29 (17.2%)	8/45 (17.8%)	
**nNRTI-NRTI**	18 (13.2%)	5/29 (17.2%)	5/45 (11.1%)	
**More than 3 drugs**	23 (16.9%)	9/29 (31.0%)	12/45 (26.7%)	
**Clinical diagnosis, *n***				NA
**N/N issues**	27 (19.8%)	17 (14.0%)	-	
**Symptomatic NCI**	9 (6.6%)	10 (8.3%)	-	
**Asymptomatic**	31 (22.8%)	42 (34.7%)	-	
**Isolated brain MRIa**	42 (30.9%)	39 (32.2%)	-	
**HIV-related syndromes**	27 (19.8%)	13 (10.7%)	-	
**CNS OIs**	-	-	38 (52.8%)	
**CNS infections**	-	-	8 (11.1%)	
**CNS cancer**	-	-	6 (8.3%)	
**Others**	-	-	22 (30.5%)	
**CSF AD biomarkers group,**				<0.001
**A**	122 (89.7%)	97 (80.2%)	43 (59.7%)	<0.001
**B**	9 (6.6%)	9 (7.4%)	8 (11.1%)	0.507
**C**	4 (2.9%)	14 (11.6%)	18 (25.0%)	<0.001
**D**	1 (0.7%)	1 (0.8%)	3 (4.2%)	0.116
**CSF tau, pg/mL**	146.1 (85.1–222.8)	120.0 (67.0–231.3)	232.8 (89.6–427.3)	0.007
**CSF ptau, pg/mL**	34.9 (27.6–45.0)	33.7 (22.3–43.2)	35.3 (24.1–44.2)	0.611
**CSF BA42, pg/mL**	891.3 (701.0–1073.8)	887.0 (673.3–1089.0)	854.9 (631.5–1076.6)	0.732
**APOE genotyping, *n***				0.440
**ε1/ε1**	0/67	1/80 (1.3%)	0/36	
**ε2/ε2**	1/67 (1.5%)	2/80 (2.5%)	0/36	
**ε2/ε3**	5/67 (7.5%)	8/80 (10.0%)	2/36 (5.6%)	
**ε3/ε3**	53/67 (79.1%)	53/80 (66.2%)	28/36 (77.8%)	
**ε3/ε4**	7/67 (10.4%)	14/80 (17.5%)	5/36 (13.9%)	
**ε4/ε4**	1/67 (1.5%)	2/80 (2.5%)	1/36 (2.8%)	
**CSF Neopterin, ng/mL**	0.73 (0.45–1.25)	2.07 (1.04–4.60)	2.51 (1.18–8.42)	<0.001
**CSF S100b, pg/mL**	117.0 (76.7–186.5)	136.9 (84.4–229.6)	146.5 (71.0–202.5)	0.311
**CSAR**	5.0 (3.9–6.6)	5.7 (4.1–7.4)	7.8 (5.2–9.6)	<0.001
**Tourtellote index**	0.05 (0.0–8.6)	6.9 (0.1–25.7)	6.7 (0.0–25.9)	<0.001
**CSF IgG synthesis, %**	0.0 (0.0–17.5)	0.0 (0.0–35.0)	0.0 (0.0–19.0)	0.158
**CSF cells, cells/mmc**	0 (0–0)	0 (0–2)	0 (0–14)	<0.001
**CSF proteins, mg/dL**	43 (33–53)	48 (39–59)	60 (43–81)	0.003
**CSF glucose, mg/dL**	59 (55–63)	50 (46–55)	54 (47–61)	<0.001

* In patients with detectable viremia and CNS confounding, the reported values refer to the on cART subgroup. Legend: CNS, central nervous system; MSM, males who have sex with other males; IDU, intravenous drug users; CSF, cerebrospinal fluid; N/N, neurological and/or neurocognitive; NCI, neurocognitive impairment; MRIa, magnetic resonance imaging abnormalities; OIs, opportunistic infections; AD, Alzheimer’s dementia; CSAR, cerebrospinal fluid-to-serum albumin ratio; NA, comparison not assessed as by definition the variable has to be different according to clinical grouping.

## Data Availability

Raw data supporting the findings of this study are available from the corresponding author M.T. on request.

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
