# Peer review of "Patterns of Cerebrospinal Fluid Alzheimer’s Dementia Biomarkers in People Living with HIV: Cross-Sectional Study on Associated Factors According to Viral Control, Neurological Confounders and Neurocognition"

_viruses, 2022, doi:10.3390/v14040753_

Round 1

Reviewer 1 Report

Each figure or table should be described in the results section by mentioning the figure/table. Overall, presentation was confusing for readers to acquire the information. All comparisons suggested by this reviewer were included in the text, but figures were not made using those. Choice of figures is misleading.    

Author Response

We have now expanded the Figures description in the result sections (see lines 197-205 for Fig.1; 208-212 for Fig.2; 259-266 for Fig.3; 270-288 plus 322-329 plus 348-350 for Fig.4; 313-318 for Fig.5).

Thanks to Rev.1 for her/his time spent at reading and improving our manuscript 

Reviewer 2 Report

The authors have addressed all the concerns I had.

Author Response

Thanks to Rev.2 for her/his time spent at reading and improving our manuscript

Reviewer 3 Report

The authors answered my critical questions well and this revised manuscript was satisfactorily modified for my comments. Thus, I am recommending acceptance for publication in the Viruses.

Author Response

Thanks to Rev.3 for her/his time spent at reading and improving our manuscript 

Round 2

Reviewer 1 Report

The aim of this study appeared to be determining the effect of HIV infection on AD. The authors approach for this goal was to use patients' plasma HIV levels as a parameter. The critical problem in the design of this study was to set aside the "HIV+ confounding" from other two groups, and not detecting the plasma HIV levels. In fact, the first two groups are "non-confounding non-viremia" and "non-confounding viremia". It is an open question if all "confounding" patients are viremia, which may leads to a quite different conclusion: plasma HIV levels and early signs of AD have high correlation. Because of the inappropriate comparison, the authors failed to achieve the goal described in the abstract.     

Author Response

Rev#1: The aim of this study appeared to be determining the effect of HIV infection on AD. The authors approach for this goal was to use patients' plasma HIV levels as a parameter. The critical problem in the design of this study was to set aside the "HIV+ confounding" from other two groups, and not detecting the plasma HIV levels. In fact, the first two groups are "non-confounding non-viremia" and "non-confounding viremia". It is an open question if all "confounding" patients are viremia, which may leads to a quite different conclusion: plasma HIV levels and early signs of AD have high correlation. Because of the inappropriate comparison, the authors failed to achieve the goal described in the abstract.     

AR: The aim of our study has been repeatedly stated in the abstract (lines 16-19: "We performed a retrospective study to analyse the distribution of cerebrospinal fluid (CSF) AD biomarkers (...) in adult PLWH (on cART with undetectable viremia, ..., with detectable viremia, ... and with central nervous system CNS disorders regardless of viremia, n=72) who underwent ..."), at the end of the background (lines 84-89: "we described CSF AD biomarkers levels and their combinations in three common clinical categories of PLWH nowadays (...) and we assessed whether several HIV-related and unrelated characteristics may differentially associate with CSF biomarkers patterns in each clinical group"), and at the beginning of methods (lines 91-95). Our study is a retrospective, observational, hypothesis-generating, prevalence study. This kind of study design can only detect associations, and cannot describe causality, so neither the primary outcome nor the secondary outcomes were “determining the effect of HIV infection on AD”, as also acknowledged in the discussion (387-388, "Our study was not designed to assess accelerated or accentuated neurodegeneration in PLWH..."). We aimed at simply describing the prevalence of the four CSF AD biomarkers patterns among the three most common and easily identifiable clinical categories nowadays (primary outcome), and at observing any potential association between these CSF patterns and HIV-related (CD4, nadir, HIV-RNA, lenght of infection, etc) and unrelated collected data (other CSF biomarkers, brain imaging, neurocognition, demographics) in each clinical category (secondary outcomes). Plasma HIV-RNA was measured and reported in the manuscript for all the included subjects, regardless of the clinical group; therefore, it is not “an open question if all HIV+ with CNS confounding are viremic”, as the answer is already reported in the results section (median plasma HIV-RNA was 2.6 Log10 cp/mL for HIV+ CNS confounding group, as reported in Tab.1). We have run non-parametric Spearman’s rank correlations between both plasma and CSF HIV-RNA and CSF biomarkers in the whole HIV+ study population, as well as in each of the three clinical categories, and none was statistically significant as reported in lines 268-269 and 321-322 ("None of the three CSF AD biomarkers correlated with any of the viro-immunological parameters") and lines 346-351 ("CSF tau correlated with ...current CD4+ T-cells count (rho -0.352, p=0.003), CSF HIV-RNA (rho 0.348, p=0.003), ...length of HIV infection (rho 0.309, p=0.009) and length of viral suppression (rho -0.457, p=0.022). Instead, CSF ptau did correlate with ...length of viral suppression (rho -0.417, p=0.038). BA42 did not correlate with any assessed variable..."). We did not report nor comment the null correlations between plasma (and CSF) HIV-RNA and CSF AD biomarkers levels in the whole HIV+ study population as it is a too much heterogeneous population and such an analysis would be meaningless; still, we analyzed it by clinical category and no association was seen. So, none of the CSF biomarkers correlate with plasma HIV-RNA, and this should not surprise, as not corroborated by previous literature.

This manuscript is a resubmission of an earlier submission. The following is a list of the peer review reports and author responses from that submission.

Round 1

Reviewer 1 Report

In this study, the authors have investigated the pattern of CSF AD biomarkers (beta amyloid 1-42, tau, phosphorylated tau) in PLWH (with 136 undetectable viremia, 121 detectable viremia, 72 CNS disorders).

There are critical points that the authors may consider.

  1. Studies of Alzheimer's cerebrospinal markers in HIV-infected people have already been published in several papers, and in terms of originality, this is the weakest point of this paper.

1) Cysique LA et al., APOE epsilon4 moderates abnormal CSF-abeta-42 levels, while neurocognitive impairment is associated with abnormal CSF tau levels in HIV+ individuals - a cross-sectional observational study. BMC Neurol. 2015 15:51.

2) de Almeida SM et al., Biomarkers of neuronal injury and amyloid metabolism in the cerebrospinal fluid of patients infected with HIV-1 subtypes B and C. J Neurovirol. 2018 24(1):28-40.

3) Lobo JD et al., CSF markers of AD-related pathology relate specifically to memory impairment in older people with HIV: a pilot study. J Neurovirol. 2022 Feb 1. doi: 10.1007/s13365-021-01048-x

  1. In this paper, the levels of tau, p-tau, amyloid 1-42 was measured using the immune-enzymatic method. The validation of this result using western blot analysis should be performed.

  1. In Table 1 and Figures, a p-value should be presented to show the significance comparison of the groups.

Reviewer 2 Report

This article entitled “Patterns of cerebrospinal fluid Alzheimer’s Dementia biomarkers in People living with HIV: cross-sectional study on 3 associated factors according to viral control, neurological confounding and neurocognition” by Trunfio et al. examined a considerable number of HIV patients to elucidate the effect of HIV infection on AD by analyzing their CSF for AD biomarkers and examining their cognitive functions. Although the topic is of great interest to neuroscientists, the study fails to demonstrate such effects due to insufficient/inappropriate analyses as described below:

Major Points

All analyses should be compared to age matched healthy patients with equivalent male/female ratio. For instance, if the authors intend to claim the abnormal correlation between [total tau] and [p-tau] etc. in HIV patients, same correlation in the non-HIV subjects should be presented as “control”.

Correlation between HIV and AD should be studied by comparing [HIV] in CSF and [total tau], [p-tau], [beta Amyloid]. In fact, comparison between detectable and undetectable plasma viremia groups does show a tendency that suggests a possible correlation.

Separating HIV patience with cognitive impairments (regardless of + or – for viremia) from the other two groups (viremia+ and viremia-) and comparing it as one independent group is inappropriate.

Minor Points

Figure 1 is not mentioned in the results.

Results should be described not by separating “detectable” and “undetectable” groups, but by comparing differences between those groups in [total tau] vs [bA] plot etc.

Reviewer 3 Report

The article by Trunfio et al., presents a cross-sectional study on factors associated with  Alzheimer’s Dementia (AD) biomarkers in PLWH. The authors retrospectively used a significantly large number of patient data to analyze the presence of β-amyloid peptide (1-42), tau, and phosphorylated tau as biomarkers of AD. With all controls, a total of 329 patients were included in the study. These patients were divided into four groups presenting different patterns. Of these 136 patients had HIV-RNA < 50 while 121 had HIV-RNA > 50. There were 84 individuals in the control group. This is an interesting study and considering that HIV-associated neurological disorders are a concern among the patient receiving the antiretroviral therapy, this study is significantly important. However, there are some points that are needed to be addressed.  

  1. The results presented in Figure showed that HIV+ (undetectable) or HIV+ (detectable) have no significant differences in 4 patterns. HIV+ CNS confounding individuals appears to be different from the other two groups. A problem here is that no p-value is provided to understand the significance of the difference.
  2. Another major problem with the manuscript that is is not clear how tau or β-amyloid peptide (1-42) or phosphorylated tau were detected. 
  3. A rationale for using APOE genotyping data is not presented, and how many subjects were included with available data (line 148).
  4. Figure 1B shows a clear correlation between β-amyloid peptide (1-42) and CSF total tau. I am not sure why the correlation was not determined here

Minor - All figures appear of low quality.